# The Chromatin Landscape around DNA Double-Strand Breaks in Yeast and Its Influence on DNA Repair Pathway Choice

**DOI:** 10.3390/ijms24043248

**Published:** 2023-02-07

**Authors:** Chiara Frigerio, Elena Di Nisio, Michela Galli, Chiara Vittoria Colombo, Rodolfo Negri, Michela Clerici

**Affiliations:** 1Department of Biotechnology and Biosciences, University of Milano-Bicocca, 20126 Milan, Italy; 2Department of Biology and Biotechnologies “C. Darwin”, Sapienza University of Rome, 00185 Rome, Italy; 3Institute of Molecular Biology and Pathology (IBPM), National Research Council (CNR) of Italy, 00185 Rome, Italy

**Keywords:** DNA double strand break, NHEJ, homologous recombination, histone phosphorylation, histone methylation, histone ubiquitylation, histone acetylation, *Saccharomyces cerevisiae*

## Abstract

DNA double-strand breaks (DSBs) are harmful DNA lesions, which elicit catastrophic consequences for genome stability if not properly repaired. DSBs can be repaired by either non-homologous end joining (NHEJ) or homologous recombination (HR). The choice between these two pathways depends on which proteins bind to the DSB ends and how their action is regulated. NHEJ initiates with the binding of the Ku complex to the DNA ends, while HR is initiated by the nucleolytic degradation of the 5′-ended DNA strands, which requires several DNA nucleases/helicases and generates single-stranded DNA overhangs. DSB repair occurs within a precisely organized chromatin environment, where the DNA is wrapped around histone octamers to form the nucleosomes. Nucleosomes impose a barrier to the DNA end processing and repair machinery. Chromatin organization around a DSB is modified to allow proper DSB repair either by the removal of entire nucleosomes, thanks to the action of chromatin remodeling factors, or by post-translational modifications of histones, thus increasing chromatin flexibility and the accessibility of repair enzymes to the DNA. Here, we review histone post-translational modifications occurring around a DSB in the yeast *Saccharomyces cerevisiae* and their role in DSB repair, with particular attention to DSB repair pathway choice.

## 1. Introduction

DNA double-strand breaks (DSBs) are highly cytotoxic DNA lesions. DSB repair failure can cause loss of chromosome fragments and cell death, while their inaccurate repair can result in chromosome rearrangements, genome instability, and ultimately cancer [1].

DNA DSBs can be repaired by multiple pathways with different levels of fidelity. The main mechanisms are non-homologous end joining (NHEJ), which results in a direct ligation of the broken DNA ends [2], and homologous recombination (HR), which is initiated by the nucleolytic processing of the 5′-terminated DSB ends (resection) followed by recombination with a homologous template, usually the sister chromatid or the homologous chromosome in diploid organisms [3]. NHEJ is considered an error-prone mechanism that can lead to the insertion or the deletion of few nucleotides when the broken ends need to be processed in order to become a suitable substrate for the DNA ligase that rejoins the DNA extremities [2]. In contrast, HR is generally error-free, even though it can have drawbacks leading to gene deletion or amplification. In addition, some homology-dependent repair pathways fix the broken ends at the cost of introducing mutations or chromosome rearrangements. These mechanisms include single-strand annealing (SSA), which repairs DSBs occurring between homologous sequences on the same chromosome and break-induced replication (BIR), which repairs one-ended DSBs that invade a homologous template and trigger extensive DNA synthesis. These repair mechanisms cause deletions and loss of heterozygosity, respectively [4,5].

The choice between DSB repair pathways with different outcomes and mutagenic potential is decisive for the maintenance of genomic integrity and it is subjected to complex regulatory mechanisms. The origin and the structural features of the DSBs, as well as the phase of the cell cycle in which they occur, are important determinants of DSB repair pathway decision. DNA end resection commits DSB repair to HR by creating 3′ single-stranded DNA (ssDNA) overhangs necessary for HR, while destroying the substrate for NHEJ repair [6,7]. During S and G2 phases of the cell cycle, resection is stimulated by the action of cyclin-dependent kinases (CDKs), thus coordinating HR with the presence of a sister chromatid, the preferred template for this kind of repair in somatic cells [5]. In contrast, NHEJ is active throughout the cell cycle, but it is predominant in G1, when resection is less efficient [8,9]. The choice among the different DSB repair pathways is also dictated by the ability of the free resected ends to search for a homologous sequence in the genome [10,11]. DNA breaks or structures that trigger homology-dependent repair can also arise during S phase at replication forks blocked by an obstacle or by replication stress. Repair of these structures is subjected to a different regulation compared to conventional DSBs [5]. In particular, the remodeling of stalled forks and/or replisome disassembly from these structures induce HR-based mechanisms in order to restore a functional replication fork [5].

The ratio between NHEJ and HR varies across phylogenetic groups. HR is generally prevalent in organisms with a small genome and with low abundance of repetitive sequences, such as budding yeast. In mammals and plants, NHEJ is the preferred pathway. On average, 75% of the DSBs that occur in mammalian cells are repaired by NHEJ, while HR repairs the remaining 25%. However, HR is always used as the preferred mechanism to repair DSBs that occur during S phase [12,13].

The response to DSBs should be considered within the context of chromatin, a highly organized structure in which the DNA is wrapped around octamers of four core histones H2A, H2B, H3, and H4 forming the nucleosomes [10]. Nucleosomes impose a barrier to the DNA repair machinery, particularly to DNA end resection, and nucleosomal organization around a DSB must be disrupted or modified to allow proper repair [14]. In addition, the position of a DSB in the genome and the chromatin structure around the DSB affect its recombination properties. Furthermore, chromatin movement on both the broken chromosome and the template increases the possibility to successfully complete HR [11,15]. Therefore, it is clear that key steps in the DSB response depend on specific chromatin modifications.

Chromatin structure around a DSB can be modified by removing entire nucleosomes, thanks to the action of chromatin remodeling factors, and/or by modifying histones, thus increasing chromatin flexibility and accessibility of repair enzymes to the DSB [10,14]. Histones are subjected to a vast array of post-translational modifications (PTMs) such as phosphorylation, acetylation, methylation, and ubiquitylation that target histone tails [16]. Histone PTMs either directly influence the overall structure of chromatin or regulate (positively or negatively) the binding of effector molecules (also called “readers”), which interact with modified histones [14,16].

An extensive work has been carried out to define the role of histones dynamics in DSB response in both the model organism *Saccharomyces cerevisiae* and in mammalian cells. Considering the evolutionarily distance between these two organisms and the extensive variations of histone marks during the cell cycle and in different cell types in mammals, one would expect that these chromatin modifications could be extremely different in the two species. In contrast, many histone PTMs and histone-modifying enzymes are well conserved, although more intricate and generally redundant pathways have evolved in metazoans compared to yeast. Taking these differences into account, we believe that the acquired knowledge of chromatin landscape and its effects on DSB repair pathway choice in yeast was and still is pilot for identifying proteins and pathways sensitive to specific histone PTMs also in mammals.

In this review, we will focus on histone PTMs and discuss recent insights into their functions in DSB repair pathway choice, mainly referring to the latest discoveries in yeast. We and others recently reviewed the findings obtained in mammals and the regulatory models which have been proposed [17,18,19,20,21]. Some reviews that explore in detail the role of chromatin remodelers in DSB repair have also been recently published [14,22].

## 2. The Cellular Response to DNA Double-Strand Breaks

DSBs elicit both DNA repair and a DNA damage checkpoint, which couples DNA repair with cell cycle progression. This DSB response is orchestrated by the phosphoinositide 3-kinase-related (PIKK) protein kinases ATM and ATR (Tel1 and Mec1, respectively, in *S. cerevisiae*), which recognize aberrant DNA structures and phosphorylate a multitude of targets involved in both repair and checkpoint pathways, thus promoting DSB repair and ensuring its completion before the cells progress through the cell cycle [6,23]. Tel1/ATM is activated by double-stranded DNA (dsDNA) ends, where it is recruited by the complex called MRX (Mre11-Rad50-Xrs2) in yeast and MRN (MRE11-RAD50-NBS1) in mammals. In contrast, Mec1/ATR activation depends on its interacting factor Ddc2/ATRIP and on the presence of long ssDNA stretches covered by the ssDNA binding protein Replication Protein A (RPA). Other proteins, such as the Dpb11/TOPBP1 scaffold, together with the 9-1-1 checkpoint clamp complex (Ddc1-Rad17-Mec3/RAD9-RAD1-HUS1) and the DNA replication/DNA damage repair proteins Dna2 (in yeast) and ETAA1 (in mammals), further stimulate Mec1/ATR activity. Once activated, Mec1/ATR and Tel1/ATM arrest the cell cycle progression through the phosphorylation of the downstream checkpoint kinases Rad53/CHK2 and Chk1, whose complete activation also requires the Rad9/53BP1 scaffold. Besides contributing to checkpoint activation, Rad9/53BP1 also regulates DSB end processing [6,24]. Mec1/ATR and Tel1/ATM target several chromatin modifiers, whose action on the modification of chromatin structure likely improves DNA repair efficacy [14,22].

The core DSB repair machinery is evolutionarily conserved from yeast to mammals, although several mammalian proteins are absent in budding yeast (e.g., DNA-PKcs in NHEJ and PARP1, BRCA1 and BRCA2 in HR). In both yeast and mammals, the first factors that bind DSB ends are the Ku70-Ku80 (Ku) heterodimer and the MRX/MRN complex. While Ku promotes the recruitment of the DNA ligase IV complex that allows NHEJ, MRX(N) initiates resection together with Sae2/CtIP through an endonucleolytic cleavage of the 5′ DSB ends, thus promoting HR [6,22]. After a brief description of the NHEJ and HR mechanisms, we will focus on DSB end resection as the main process determining the DSB repair pathway choice.

### 2.1. Non-Homologous End Joining

During NHEJ, DNA ends with little or no complementary base pairing are kept together by a multiprotein synaptic complex and they are directly rejoined by a DNA ligase (Figure 1a). A perfect re-ligation of the broken ends is referred to as canonical non-homologous end joining (c-NHEJ), it is error free and completely depends on the Ku complex. c-NHEJ is initiated by the binding of Ku to the DNA ends. This binding protects the DSB ends from degradation and promotes the recruitment of the DNA ligase IV (Dnl4/Lig4 in yeast), which accomplishes the end joining reaction, together with its associated proteins Lif1/XRCC4 and Nej1/XLF [2,25].

In both yeast and mammals, alternative non-homologous end joining (alt-NHEJ) events are observed in the absence of Ku. Most of these events, which are also referred to as microhomology-mediated end joining (MMEJ), rely on very short microhomology sequences that are exposed by limited resection of DSB ends. These sequences anneal to each other and the subsequent re-ligation generates small insertions/deletions. Therefore, alt-NHEJ is highly mutagenic [2,25].

### 2.2. Homologous Recombination

HR can be considered as a collection of alternative processes optimized to properly repair each DSB, based on its structural features and origin (Figure 1b,c) [3,4]. Almost all HR mechanisms are characterized by three main steps, which are crucial to define the different repair pathways: (i) the initiation, which consists in the creation of 3′-ended ssDNA tails by resection, (ii) the homologous DNA pairing, which requires the search for sequence complementarity and the formation of a heteroduplex intermediate, and (iii) the resolution of the intermediate, which determines the HR outcomes.

HR is initiated by resection of both DSB ends to create 3′-ended ssDNA tails in a two-step process that requires the concerted action of helicases and nucleases. The first resection step (short-range resection) requires MRX/MRN and Sae2/CtIP, whose action is particularly important to process DNA ends with protein adducts or “dirty” ends, while it is dispensable for resection of “clean” DSB ends, as those induced by endonucleases [26]. Mre11 possesses both 3′-5′ dsDNA exonuclease and ssDNA endonuclease activities, while Rad50 has an ATPase domain that regulates the activity of the complex [27]. Indeed, ATP binding and hydrolysis by Rad50 regulates the MRX/MRN binding to the DNA ends, the plasticity of the complex and its nuclease activity (reviewed in [22]). The endonuclease activity of Mre11 is also stimulated by Sae2/CtIP and it creates an internal nick at ~20–50 nt from the 5′ ends, followed by the 3′-5′ exonucleolytic degradation exerted by Mre11 itself back toward the DNA ends [28,29]. This MRX action contributes to displace tightly bound proteins from DSB ends, thus creating a short 3′ overhang that provides an entry site for the long-range resection nucleases Exo1 and Dna2. While Exo1 possesses 5′-3′ exonuclease activity capable to release single nucleotides from dsDNA ends [30], Dna2 resection activity requires that the RecQ helicase Sgs1/BLM unwinds the duplex DNA [31,32]. Exo1 and Dna2 can resect thousands of nucleotides in length creating long 3′-ended ssDNA tails, which are rapidly coated by RPA.

After DSB end resection, the most common HR mechanisms require strand pairing and strand invasion, which strictly depend on the RecA/Rad51/Dmc1 family of recombinases that form a nucleoprotein filament on ssDNA. Rad52 together with a set of Rad51 paralogs, replaces RPA with Rad51 in yeast, while in mammals the main loader of RAD51 is BRCA2, along with several RAD51 paralogs [3,4]. Once the Rad51 nucleoprotein filament is formed, it has the capacity to scan the entire genome, searching for a region of homology within dsDNA and to promote base pairing and strand exchange, thus leading to the formation of a displacement loop (D-loop). As well as resection, another rate-limiting step in HR is the search in the genome for a homologous sequence to use as a template.

For a long time, scientists interested in HR mechanisms have wondered about this “search problem”, thus showing that repair of a DSB is influenced by the proximity between the damaged and the template sequences. This homology search could also be facilitated by DNA damage-stimulated changes in chromosome movement (reviewed in [11,15]).

After the D-loop formation, HR can proceed via the canonical pathway that involves the formation of a double Holliday junction, whose resolution can generate either crossover or non-crossover products (Figure 1b, center) [3]. Alternatively, if the D-loop intermediate is not stabilized by a second-end capture, the displaced nascent strand can re-anneal to the template, thus generating only non-crossover products. This mechanism is termed synthesis-dependent strand annealing (SDSA) and it is the predominant HR pathway in mitotic cells (Figure 1b, left). In contrast, stabilization of the D-loop in combination with the failure of second-end capture leads to BIR, which triggers extensive DNA synthesis, potentially until the end of the template chromosome (Figure 1b, right).

A specialized homology-dependent DSB repair pathway which does not require strand invasion and D-loop formation is the SSA. This mechanism relies on the annealing between two homologous sequences flanking a DSB and leads to the deletion of one of the repeats and of the intervening region (Figure 1c) [5].

### 2.3. The Regulation of DNA End Resection and the Choice between NHEJ and HR

DSB end resection is subjected to multiple layers of regulation, which rely on two main molecular events: the regulation of the nuclease activity and the modulation of the accessibility/persistence of these enzymes on the broken DNA ends [6,7,22,24]. According to their effects on resection, we can distinguish between positive and negative events of regulation. Positive regulation of resection is exerted by the action of the CDKs (Cdk1 in *S. cerevisiae*) [8,9]. Instead, negative regulation can be ascribed to proteins that act as a barrier to resection initiation or progression, such as the Ku complex and the Rad9/53BP1checkpoint adaptor [24].

Inefficient resection in G1 is due to both low CDK activity and the binding to the DNA ends of Ku, which blocks the access of Exo1 to DNA [8,9,33,34,35,36,37]. In the absence of Ku, CDK-independent resection occurs in regions proximal to the DSB ends, suggesting that CDK activity could inhibit Ku. Conversely, extensive DSB end resection is enhanced by high CDK activity, but not by Ku absence, thus indicating that CDK activity promotes resection also independently of Ku [33]. Indeed, CDK promotes both short- and long-range resection by phosphorylating and activating Sae2 and Dna2, respectively [38,39]. The CDK-mediated Sae2 phosphorylation stimulates Mre11 endonuclease activity [35,37,40]. In both yeast and mammals, Sae2/CtIP is also subjected to phosphorylation by the checkpoint kinases Mec1/ATR and Tel1/ATM [41]. In yeast cells, this phosphorylation promotes Sae2-Rad50 interaction, which stimulates the endonucleolytic activity of Mre11, while in mammals NBS1 interacts with phosphorylated CtIP and mediates MRE11 activity stimulation [42,43,44,45].

The Mre11-mediated endo/exonucleolytic processing of DNA ends is important to remove chemical modifications or proteins that can hamper the access of the long-range resection nucleases. While Ku specifically inhibits Exo1, Rad9/53BP1 counteracts the action of Sgs1-Dna2 by limiting Sgs1 association to the DNA ends [46,47], with only minor effects on the Exo1-dependent resection [48]. Rad9 interacts with modified histones H3 and H2A and its recruitment to damaged DNA involves both histone-dependent and histone-independent mechanisms. Rad9 interacts with methylated histone H3 through its Tudor domain independently of DNA damage [49,50,51] and it is further recruited to damaged DNA through an interaction between the BRCT domain and the phosphorylated histone H2A (Figure 2) [52,53,54,55]. Rad9 association to DSB also requires the multi-BRCT domain protein Dpb11 [56,57], while it is subjected to negative regulation by both the chromatin remodeler Fun30 (SMARCAD1 in mammals) and the scaffold protein complex Slx4-Rtt107 [6,24]. This fine-tuned control of Rad9 association to the DSBs is likely important to generate enough ssDNA to guarantee homology-directed DSB repair. Moreover, it limits excessive resection, which can lead to inefficient repair and chromosome instability.

In addition to Rad9, the checkpoint 9-1-1 complex has been recently found to regulate resection in both positive and negative manners. 9-1-1 restricts the MRX access to the broken ends and limits the Sgs1-Dna2-dependent long-range resection through the stabilization of Rad9 binding to chromatin. On the other hand, it enhances the Exo1-mediated resection through undefined mechanisms [48,58]. Finally, the recombination protein Rad52 was found to negatively regulate long-range resection independently of Rad9 by limiting the translocation of the helicase Sgs1 on ssDNA [59].

## 3. Chromatin Structure and Histone Modifications

In eukaryotes, chromatin is characterized by the presence of nucleosomes, which are constituted by ∼146 base pairs of DNA wrapped approximately twice around two copies of H2A–H2B and H3–H4 dimers (histone core). Chromatin also comprises additional proteins, including the linker histone H1, which is responsible for higher-order chromatin structure. Histone tails protrude from the nucleosomes and are subjected to a vast array of PTMs, which regulate transcription and other DNA metabolism processes, including DNA repair [60]. Nucleosome stability also varies due to the incorporation of non-canonical histone variants [10]. The occurrence of a DSB causes deep changes in the chromatin landscape of a large domain of the broken chromosome. The maintenance of genome integrity is affected by these extensive alterations in the chromatin components that contribute to DSB repair.

Nucleosomes form a barrier to resection by limiting the action of nucleases. In fact, Mre11 preferentially cleaves nucleosome-free DNA [29,61], Exo1 is unable to resect a nucleosome-rich substrate in vitro, while Sgs1-Dna2 can process DNA wrapped around nucleosomes only when enough nucleosome-free DNA is present [62]. A recent study has described how the APE1 endonuclease can cleave an apurinic/apyrimidinic (AP) site in a nucleosome and trigger base-excision repair depending on the AP site position in the nucleosome [63]. Cryo-electron microscopy analyses have revealed that when the AP site is exposed on the nucleosome surface, APE1 induces a distortion of the DNA without structural rearrangements of the histone core and bends the nucleosomal DNA in the APE1 active site. In contrast, the extensive interactions between nucleosomal DNA and the histone octamer prevent AP site cleavage when this site is occluded into the nucleosome [63]. Additional activities help nucleases to overcome the chromatin barrier. Exo1-mediated resection is enhanced when H2A is replaced with the less abundant H2A.Z variant, which decreases nucleosome stability. Furthermore, acetylation of H3 K56 increases spontaneous unwrapping of nucleosomal DNA and enhances APE1 cleavage at occluded nucleosomal AP sites [63,64]. Taken together, these findings suggest that nucleosome removal or repositioning is important to allow efficient nuclease action [62]. Indeed, several chromatin remodelers have been found to promote resection (reviewed in [14]).

Histone PTMs participate in the DSB response by acting at different levels and supporting different pathways for DNA repair. These PTMs affect chromatin structure in different ways: they modify histone-histone or histone-DNA interactions, or they provide binding sites for other factors. The modification of histone-histone or histone-DNA interactions may alter nucleosome stability or long-range contacts among nucleosomes. In addition, chromatin chaperons and remodeling enzymes are recruited to the damaged DNA through the interaction with specific histone PTMs and they can either disassemble or slide nucleosomes on the DNA [10,14].

Histone phosphorylation, methylation, ubiquitylation, and acetylation have been clearly involved in DSB repair in both yeast and mammals [10,14,18,65,66,67]. How these PTMs modify the chromatin landscape around a DSB and influence DSB repair pathway choice in yeast is detailed below.

## 4. Phosphorylation of Histone H2A

Phosphate groups are added to histones by protein kinases on threonine, tyrosine and serine residues. This modification reduces the overall charge of the histones and increases the availability of the DNA to be bound by specific proteins [60]. The role of histone phosphorylation in DSB repair has been extensively explored and at least partially defined. In yeast, S122, T126, and S129 residues on the H2A C-terminal tail undergo DNA damage-induced phosphorylation events, which have been linked to meiotic recombination, repair of fragile sites with CAG/CTG repeats and DSB repair, respectively [68,69]. In yeast, H2B T129 is also phosphorylated in a Mec1- and Tel1-dependent manner, and it parallels the function of phosphorylated H2A in DNA repair [70], but this modification is not conserved in mammals. Recently, a checkpoint-dependent phosphorylation on the H2A N-terminus (H2A S15) has also been described [71,72]. This phosphorylation, along with that of S129, has been involved in DSB repair pathway choice.

### 4.1. H2A S129

The most characterized histone PTM in DSB repair is the phosphorylation of H2A S129, termed γ-H2A, which corresponds to S139 residue on mammalian H2A.X variant (γ-H2A.X—we will use γ-H2A for both). Both mammalian H2A.X S139 and yeast H2A S129 are rapidly phosphorylated in the surroundings of DSBs by the checkpoint kinases Mec1/ATR and Tel1/ATM (Figure 3) and are therefore largely used as DSB markers [53,73]. In yeast, γ-H2A spreads approximately 50 kb on both sides of the DSB about 30 min after DSB occurrence [53]. Tel1 plays a major role in H2A phosphorylation in G1 and close to the DSB ends [53], while Mec1 phosphorylates H2A to greater distances from the DSB ends, reflecting the resection kinetics (about 4 kb/hour) [74]. Unlike Tel1, Mec1 is also capable to spread γ-H2A in trans on an unbroken chromosome kept in proximity of the damaged one [74,75,76].

Although H2A is phosphorylated by the checkpoint kinases, it is not required for the checkpoint-mediated cell cycle arrest. Rather, it is important for DSB repair [77,78,79]. Indeed, yeast cells in which the two H2A encoding genes *HTA1* and *HTA2* are mutated to produce a non-phosphorylatable H2A-S129A variant are hypersensitive to the treatment with DSB-inducing agents [52,77]. γ-H2A regulates DSB repair pathway choice and promotes HR by three different mechanisms: it regulates DNA end resection, it promotes cohesion between sister chromatids, and it increases chromosome mobility, thus favoring homology searching.

γ-H2A likely exerts both positive and negative control on resection. Resection was found to be accelerated in *H2A-S129A* mutant cells compared to wild type ones [80,81]. Indeed, γ-H2A increases Rad9 binding to the DSB ends through an interaction between phosphorylated S129 and the Rad9 tandem-BRCT domain (Figure 2), thus inhibiting resection [52,54,55,74]. Despite the fast kinetics, resection in *H2A-S129A* mutant cells is inefficient, indicating that γ-H2A also plays a positive role in DNA end resection. In fact, γ-H2A acts as a docking site for chromatin remodeling enzymes. γ-H2A physically interacts with Arp4, a subunit of different chromatin remodeling complexes, including the INO80, the SWR1 and the NuA4 complexes [82,83]. How the NuA4 histone acetyltransferase complex regulates DSB repair is discussed below. Concerning the INO80 complex, after it has been recruited to broken DNA ends, it stimulates HR by promoting both resection and presynaptic nucleofilament formation [82,84]. Indeed, INO80 evicts nucleosomes in the surroundings of the DSB, thus creating a nucleosome-free DNA region that favors the recruitment of resection nucleases [82]. In addition, INO80 has been proposed to promote Rad51 recruitment by evicting H2A.Z-containing nucleosomes, which counteract nucleofilament formation [84]. The SWR1 complex promotes the opposite reaction, as it exchanges H2A-H2B dimers with the less stable H2A.Z-H2B dimers, in an ATP-dependent manner [85]. Although INO80, SWR1, and H2A.Z are clearly involved in the regulation of resection and other steps in the HR repair mechanism, the exact functions of these factors are still under investigation.

γ-H2A is also involved in HR events downstream of the DSB ends processing. It promotes de novo recruitment to the DSBs of cohesin, a multiprotein, ring-shaped complex that tethers sister chromatids together [86,87,88]. Cohesin is normally loaded onto the DNA during the S-phase of the cell cycle and it persists until the onset of anaphase in order to ensure equal segregation of genetic material to daughter cells. In addition to this canonical role, cohesin also plays important functions in the transcription and in the response to DNA damage [89]. Indeed, the formation of a DSB in the S or the G2/M phase of the cell cycle is sufficient to induce cohesin enrichment in the surroundings of the DSB [87]. Moreover, while resection seems to normally proceed in the absence of cohesin [86], pulsed-field gel electrophoresis analyses revealed the persistence of chromosome damage after ionizing radiation (IR) exposure in cohesin-depleted cells, suggesting that cohesin is required for proper DSB repair [87,88].

Finally, γ-H2A, together with Rad51, is involved in chromosome mobility and homology search in yeast and likely in mammals [90,91,92]. The negative charges resulting from H2A phosphorylation have been proposed to create repulsive forces that can regulate chromatin stiffening and increase chromosome mobility [93,94]. Accordingly, *H2A-S129E* mutation, which mimics constitutive H2A phosphorylation, was found to increase chromosome mobility also in the absence of DNA damage. Furthermore, increased chromatin dynamics in *H2A-S129E* cells correlated with improved DSB repair. Somehow unexpectedly, NHEJ was found to be slightly enhanced in these mutant cells [95]. As the lack of γ-H2A accelerates resection [80,81], a mutation that mimics constitutive H2A phosphorylation could delay resection initiation, thus allowing the cells more time to complete NHEJ. Subsequent analyses revealed that the global chromosome mobility induced by γ-H2A is essential for HR repair of DSBs occurring far from the centromere, while it is dispensable when the DSB is generated close to the centromere [96].

Taken together, these findings suggest that γ-H2A promotes HR by coupling ssDNA generation with the research of a template for homology-driven repair. In fact, γ-H2A favors the tethering of DNA molecules and facilitates homology search, while it concomitantly modulates the resection rate. Both positive and negative regulation of resection mediated by γ-H2A is likely important to promote HR. Indeed, while chromatin remodeling allows resection to easily proceed beyond nucleosome-organized DNA regions, the recruited proteins that act as a barrier to end resection prevent excessive ssDNA generation, which could interfere with the completion of HR [97].

Findings obtained in yeast concerning the role of γ-H2A in recruiting Rad9 have led to the exploration of whether the same regulation takes place in mammalian cells. Although mammalian γ-H2A was found to directly bind the BRCT2 domain of 53BP1 in vitro [98], it does not directly recruit 53BP1 to the DSBs. However, γ-H2A is linked to the control of 53BP1 recruitment to the DSB ends through a complex regulatory circuit. γ-H2A recruits the adaptor MDC1 to damaged DNA through a direct interaction with MDC1 BRCT2 domain. MDC1, in turn, recruits several proteins, including the E3 ubiquitin ligases RNF8 and RNF168, which catalyze the ubiquitylation of K13 and K15 residues of H2A histone and the subsequent recruitment of 53BP1 and other repair factors [19]. The direct interaction between γ-H2A and 53BP1 was proposed to increase the amount of 53BP1 on the DNA, to sustain the activation of the DNA damage response and to facilitate DSB repair in heterochromatin in G1 [98]. Although the involved molecular mechanisms are more intricate in mammals than in yeast, in both organisms γ-H2A participates in the regulation of the amount of Rad9/53BP1 at the DSB ends, which is an important determinant of DSB repair pathway choice.

Despite studies on chromatin mobility in mammalian cells have yielded conflicting results, recent findings suggest that γ-H2A is also involved in damage-induced chromatin mobility and that increased chromatin movements stimulate HR [90,91,92,99]. However, additional studies are required to better define these mechanisms.

### 4.2. H2A S15

H2A S15 has been recently found to be phosphorylated by Mec1 in the presence of DNA damage over a large domain of chromatin around the DSB [71,72]. H2A S15 phosphorylation occurs independently of S129 modification, and it appears to positively regulate resection. Indeed, the H2A-S15E phospho-mimicking variant accelerates resection and increases RPA binding in the surroundings of a DSB [72]. Although H2A S15 phosphorylation appears to modulate the interaction of H2A with Rad9 in vitro, Rad9 binding to the DSB is not affected by *H2A-S15A* or *H2A-S15E* alleles in vivo. However, in these mutant cells, low levels of chromatin acetylation were found near the break, suggesting that H2A S15 phosphorylation could regulate the recruitment of acetylases or other chromatin remodelers at the damage site [72]. Interestingly, in mammals, S15 is replaced by a lysine residue, which is subjected to different PTMs that regulate resection [100,101]. In particular, H2A K15 monoubiquitylation promotes the recruitment of both 53BP1 and the BRCA1-BARD1 heterodimer. While 53BP1 promotes NHEJ, BRCA1-BARD1 triggers HR and antagonizes 53BP1 accumulation [19]. The predominance of 53BP1 or BRCA1-BARD1 (and therefore of NHEJ or HR) is a function of the cell cycle phase and depends on other histone PTMs that occur independently of DNA damage. Indeed, the Tudor domain of 53BP1 interacts with mono- or di-methylated lysine 20 on H4 histone, a histone mark that accumulates at chromatin in G1. After DSB formation in G1, 53BP1 is further enriched, thus promoting NHEJ [102]. In post-replicative chromatin, demethylation of H4 K20 allows an efficient recruitment of BRCA1-BARD1, which antagonizes 53BP1 accumulation at the DSB ends and promotes HR [19,102,103]. Although additional work will be required to determine how H2A S15 phosphorylation regulates resection, these findings raise the possibility that yeast H2A S15 might be the functional homolog of mammalian H2A K15 in resection modulation.

## 5. Methylation of Histone H3

Methylation occurs on histone lysine and arginine residues thanks to histone methyl transferases (HMTs), which use S-adenosyl-L-methionine as a substrate. This modification primarily acts through signal transduction by recruiting specific factors, but it also affects protein–DNA interactions through the alteration of histones charges [60]. Strong evidence involves histone H3 tail lysines methylation in damage signaling and repair. In yeast, the H3 residues involved in the DNA damage response are K4, K36, and K79.

### 5.1. H3 K4

H3 K4 methylation is involved in transcription regulation, and it is expected to be highly regulated around DSBs, in order to coordinate DNA transcription and repair machineries [104]. In budding yeast, tri-methylation of lysine 4 on histone H3 (H3-K4me3) is involved in maintaining genome stability [66]. The H3-K4me3 mark becomes detectable on newly created DSBs and cells that cannot methylate H3 K4 display a defect in DSB repair by NHEJ, associated with a defect of Ku recruitment to damaged DNA [66]. In yeast, Set1 is the unique HMT responsible for H3 K4 methylation (Figure 4) and it is recruited to the DSBs by interaction with the RSC nucleosome remodeling complex. Indeed, Set1, as well as H3-K4me3, accumulates at the DSB induced by the homothallic switching endonuclease (HO) [105].

Similar observations have been reported in mammalian systems, in which H3-K4me3 appears at DSBs [105,106]. Mammalian cells lacking H3 histone methylation display a significant decrease in DSB repair by NHEJ and a decreased viability in the presence of replication stress [106]. On the other hand, in mammals, there are also specific histone demethylases (HDMs) committed to H3-K4me3 demethylation, which are redistributed into the nucleus after exposure to IR and are highly enriched at the damaged sites [106]. Two of these demethylases are KDM5A and KDM5B, which are required for efficient HR and NHEJ by promoting the recruitment of BRCA1 and Ku70 to IR-induced breaks [18,106,107]. These findings suggest that in the mammalian system, a complex interplay between H3 K4 methylation and demethylation is required to regulate DSB repair.

In yeast, no evidence has been reported on a requirement for Jhd2, the unique H3 K4 HDM, for DSB repair. Indeed, while *set1*Δ strains show hypersensitivity to gamma radiation [50], *jhd2*Δ strains do not, although a dynamic equilibrium between the modification activities carried out by these two enzymes cannot be ruled out. In mammals, HDMs interact with several other proteins with transcription repressing and/or remodeling activities. Therefore, it is still not clear if the demethylating activity is directly responsible for DSB repair promotion or rather the HDM capability of recruitment of corepressors/coregulators is involved. This aspect could differentiate the complex mammalian regulatory circuit [18] from the simplified yeast version, which is summarized in Figure 4.

### 5.2. H3 K36

Links between histone H3 K36 methylation and DSB repair have been identified in budding yeast [108,109]. Indeed, the deletion of *SET2* gene, which encodes for the unique HMT responsible for the methylation of this residue in *S. cerevisiae* (Figure 4), causes hypersensitivity to X-ray radiation [110] and genotoxic drugs as doxorubicin [111]. More recent work in fission yeast specifically supports a role for H3 K36 methylation in promoting efficient NHEJ [112]. In fact, Set2-dependent H3 K36 methylation limits chromatin accessibility, reduces resection, and promotes NHEJ, while antagonistic Gcn5-dependent H3 K36 acetylation increases chromatin accessibility, stimulates resection, and promotes HR. Accordingly, loss of Set2 increases the acetylation of H3 K36, chromatin accessibility and resection, while Gcn5 loss results in the opposite phenotypes following DSB induction [112]. Consistently, H3 K36 modification is cell cycle regulated, with Set2-dependent H3 K36 methylation peaking in G1 when NHEJ occurs, while Gcn5-dependent H3 K36 acetylation is predominant in S/G2 when HR prevails. These findings support a model in which regulation of DSB repair pathway choice depends on a H3 K36 chromatin switch. Direct evidence of a role of the H3 K36 specific demethylase Jhd1 in DSB repair is lacking, although its deletion confers sensitivity to chemicals [113].

In mammalian cells, H3 K36 di-methylation is rapidly induced both globally and locally after IR treatment or DSB generation [114,115]. Di-methylation of H3 K36 improves the association of early DNA repair components, including Ku, to the DSB ends and enhances DSB repair. The di-methylation of H3 K36 residue near the DSB is directly performed by the DNA repair protein Metnase (SETMAR), which has a SET histone methyl transferase domain and is also involved in NHEJ [114]. Recently, several studies have also revealed a function of H3-K36me3 in RAD51 recruitment in active transcription-associated HR [105,116]. Once again, in the mammalian system the regulation appears to be more intricate than in yeast, being the result of a complex interplay involving several HTMs and HDMs and their relative interactors [18].

### 5.3. H3 K79

H3 K79 methylation has been involved in the regulation of telomeric silencing, cellular development, cell-cycle checkpoint, DNA repair, and transcription [117]. The evolutionarily conserved enzyme Dot1/DOT1L is the only responsible for all forms of H3 K79 methylation (mono-, di- and tri-methylation) in eukaryotes (Figure 4). In *S. cerevisiae*, deletion of *DOT1* leads to increased sensitivity to X- and gamma-rays radiation [50] and to chemicals [113]. H3 K79 methylation is accurately regulated by a crosstalk with other histone modifications, such as H3 K4 methylation and H2B ubiquitylation (Figure 2) [117]. The tandem Tudor domain of the mammalian 53BP1 protein specifically binds to methylated H3 K79 in vitro. This interaction is required for recruiting 53BP1 to the DSB ends, while its association is inhibited by DOT1L silencing. However, the methylation level of H3 K79 does not significantly increase upon induction of DNA damage. Therefore, 53BP1 could indirectly detect the DSBs, thanks to chromatin structure changes, which expose 53BP1 binding sites [118]. 53BP1 associated to the damage site could, in turn, recruit additional proteins to activate the checkpoint response. Studies in budding yeast indicate that the Tudor domain of Rad9 interacts with the methylated H3 K79 independently of DNA damage (Figure 2) and suggest that this interaction inhibits ssDNA production, thus representing a functional and physical barrier to DNA end processing at DSBs [65]. In addition, the H3 K79 di-methylation is involved in Rad51 foci formation [119]. The putative role of yeast Dot1 in determining the selection of DSB repair pathways is summarized in Figure 4.

## 6. Ubiquitylation of Histones H2A, H2B and H1

Histone ubiquitylation plays a critical role in DNA damage repair, although the consequences of this modification have been only partially defined at a molecular level [20]. Histone ubiquitylation is dynamically controlled by three enzymatic activities (exerted by E1 ubiquitin activating enzyme, E2 ubiquitin conjugating enzyme and E3 ubiquitin ligase) that promote the covalent attachment of ubiquitin to lysine residues of the target proteins, and deubiquitylation enzymes (DUBs), which reverse ubiquitin binding from the modified proteins [120]. Ubiquitylation of histone residues could exert two different functions: it could either modify histone structure/interactions or trigger histone loss. The most characterized histone ubiquitylation events implicated in the DNA damage response are the H2A ubiquitylation in mammalian cells, and the mono-ubiquitylation of H2B and of the linker histone H1 in both yeast and mammals.

H2A ubiquitylation plays a key role in orchestrating DSB repair pathway choice in mammalian cells (reviewed in [20]). As also described above, ATM-dependent ubiquitylation of H2A K13 and K15 residues by the RNF168 E3 ligase occurs at the DSBs. These histone modifications promote the association of 53BP1, which inhibits resection and allows NHEJ mainly in G1, when 53BP1 is further enriched at chromatin by the methylation of H4 K20. RNF168 also promotes the assembly of DNA repair proteins such as BRCA1, RAD18, and RAP80 at damaged chromatin. Finally, ubiquitylation of the H2A C-terminal tail is important for downstream events in HR. In fact, ubiquitylation of the H2A K125, K127, or K129 residues by the BARD1 E3 ligase complex stimulates HR, possibly through the recruitment of the chromatin remodeling factor SMARCAD1 and the USP48 DUB [20]. A similar regulation has not been identified in yeast. Furthermore, human RNF168 is unable to transfer ubiquitin to the yeast H2A variant in vitro [121], suggesting that this regulatory circuit is specific for metazoans.

### 6.1. H2B K123

In response to DNA damage, H2B is mono-ubiquitylated on K123 residue in budding yeast (K120 in mammals) by the Bre1-Rad6 E2-E3 ubiquitin ligase complex (Figure 5), orthologue of the mammalian RNF20/40 complex [122,123,124]. Consistent with a role of this histone modification in DSB response, cells lacking Bre1 or RNF20 showed hypersensitivity to IR and to DNA damaging agents [110,125]. Furthermore, Bre1 is probably recruited to the DSB ends through a direct interaction with RPA, which is associated to ssDNA. Once recruited, Bre1 stimulates local H2B ubiquitylation, which promotes Rad51 loading and the subsequent HR repair [126,127]. Similarly, the RNF20/40 complex is recruited to the DSB ends in mammalian cells and it promotes HR by stimulating chromatin relaxation and recruitment of repair factors [119,125,128,129].

An additional role of histone H2B ubiquitylation in regulating DSB end resection was documented in the fission yeast *Schizosaccharomyces pombe*. H2B K119 residue (corresponding to H2B K123 in *S. cerevisiae*) is mono-ubiquitylated in response to DSBs [130] thanks to the coordinated action of two different complexes: the HULC-Rhp6 complex (orthologue of *S. cerevisiae* Bre1-Rad6) and the conserved Cullin4-DDB1 ubiquitin ligase complex, in association with Wdr70 protein. The HULC-Rhp6 complex is sufficient to catalyze H2B K119 ubiquitylation in a region proximal to the DSB ends, while it requires the action of Cullin4-DDB1-Wdr70 to stimulate H2B K119 ubiquitylation distal to the DNA ends [130]. The lack of either Wdr70 protein or H2B K119 ubiquitylation delays resection by causing both the accumulation of the resection inhibitor Crb2 (Rad9/53BP1 orthologue) at the damaged site and a defect in Exo1 association to the DSB ends. This indicates that H2B K119 ubiquitylation promotes resection [130], but how exactly this modification facilitates Exo1 recruitment to the DNA ends is still unknown.

H2B ubiquitylation is also at the center of a complex crosstalk among different kinds of histone PTMs and it promotes the recruitment of histone modifying enzymes. In both yeast and mammals, H2B K123/H2B K120 mono-ubiquitylation promotes the methylation of H3 K4 by the Set1/COMPASS complex, methylation of H3 K36 by Set2 or Metnase and methylation of H3 K79 by Dot1/DOT1L (Figure 2) [131]. Finally, H2B ubiquitylation has been involved in the regulation of several ATP-dependent chromatin remodelers [132]. One of these remodelers is the evolutionarily conserved Chd1 protein, whose remodeling activity is stimulated by H2B ubiquitylation in vitro [133]. Interestingly, it has recently been demonstrated that Chd1 stimulates HR and enhances both short- and long-range resection by promoting the association of MRX and Exo1 to the DSB ends [134]. Since Chd1 couples ATP hydrolysis with the reduction of histone occupancy near the DSB ends and all Chd1 functions in the DSB response require its ATPase activity, Chd1 has been proposed to facilitate MRX and Exo1 processing activities by opening the chromatin structure around the DSB [134]. Altogether, these findings suggest that H2B ubiquitylation contributes to regulate HR both by favoring Rad51 nucleofilament formation and by modulating resection progression, in association with other histone PTMs and chromatin remodeling enzyme activities (Figure 5).

### 6.2. H1 K16

Histone H1 mediates higher-order chromatin folding in metazoans and it has a well-established role in chromatin organization during DNA repair [135]. RNF20/40 complex, together with the ubiquitin ligase RNF8, mediates H1 ubiquitylation, which promotes the recruitment of RNF168 to the DSB ends. RNF168, in turn, ubiquitylates H2A K13 and K15 residues, thus promoting the subsequent recruitment of repair enzymes [20,136,137,138].

Like H1 in mammals, Hho1 linker histone in budding yeast is important for DSB repair, although its exact role is still unclear. *HHO1* inactivation was found to increase the resistance to DNA damage of mutants with defective NHEJ, but not of recombination mutants, suggesting that Hho1 may specifically inhibit HR [139]. Interestingly, a recent report has identified phosphorylation events on Hho1 S65, S173 and S174 residues that seem to stimulate DSB repair by HR, particularly when the homologous sequences involved in recombination are far from each other [140]. These findings suggest a crosstalk between ubiquitylation and phosphorylation events in modulating the action of Hho1.

Hho1 was recently found to be ubiquitylated on K16 in cells treated with zeocin and to be degraded because of the activation of the DNA damage response [141]. A genome-wide proteomic analysis upon yeast chromatin before and after zeocin-induced DNA damage showed that ubiquitin ligases and proteasome subunits are enriched on damaged chromatin and they contribute to the depletion of histones, including Hho1 [141]. Rad6, Bre1, Pep5, Rsp5, and Ufd4 ubiquitin ligases are enriched at damaged chromatin in a INO80-dependent manner and contribute to histone loss during the DSB response [10,141]. Inactivation of these ubiquitin ligases impairs long-range resection, while deletion of *HHO1* enhances it, suggesting that Hho1 removal in the proximity of the DSB is important to allow resection [141]. Taken together, these findings suggest that DSB-induced Hho1 ubiquitylation allows its degradation, thus increasing chromatin accessibility to repair factors and promoting strand invasion during HR.

## 7. Acetylation of Histones H2, H3 and H4

Acetyl groups are added to the ε-amine of lysine side chains by histone acetyltransferases (HATs), using acetyl-CoA as a donor. Lysine acetylation partially causes charges neutralization and enhances chromatin accessibility, thus increasing protein binding to the DNA [60]. A dynamic regulation of histone acetylation through HATs and histone deacetylases (HDACs) promotes the activation of the DNA damage response, especially of DSB repair. Histone acetylation generally increases chromatin unwinding and favors DSB repair, while a requirement for HDAC complexes to perform efficient NHEJ has also been reported [108,142]. In yeast, the most characterized acetylation events linked to DSB repair pathway choice occur at H2A and H4 histone tails, while H3 acetylation is associated with DSB repair mainly during DNA replication.

### 7.1. H2A and H4

In yeast, the N-terminal tails of H4 and H2A histones are acetylated by the essential multi-subunit NuA4 HAT complex (TIP60/p400 complex in mammals), whose catalytic subunit is Esa1 [143,144,145,146]. Either temperature-sensitive mutations in *ESA1* gene or point mutations in acetylatable H4 and H2A lysine residues cause hypersensitivity to DNA damaging agents, indicating that both NuA4 and histone acetylation are important for DSB repair (Figure 6) [147].

NuA4 is recruited to the DSB ends by the MRX complex. Through the acetylation of both histones and non-histone proteins, NuA4 promotes resection and HR, while it inhibits NHEJ [72]. Acetylation of Nej1 and Ku80 by NuA4 was found to reduce NHEJ [67]. On the other hand, the interaction between γ-H2A and the Arp4 subunit of NuA4 in the proximity of the DSBs, regulates the recruitment of ATP-dependent remodelers (INO80, SWI/SNF and RSC) that, in turn, stimulate resection [20,83,148,149,150]. Interestingly, both NuA4 and an increased presence of acetylated H4 were also detected at the donor sequences during HR, suggesting a role for H4 acetylation in strand invasion and D-loop formation [67].

The SAGA complex is another HAT that contributes to NuA4 recruitment to the DSBs and participates in resection stimulation [67]. SAGA acetylates H3 and H2B tails through its catalytic subunit Gcn5 [151]. Depletion of both Esa1 and Gcn5 causes loss of histone acetylation and an increased nucleosome occupancy near the DSB, while the recruitment of SWI/SNF, RSC and INO80 complexes is greatly reduced [67]. These findings indicate that NuA4 and SAGA contribute to channel DSB repair into HR pathways both by acetylating and inhibiting key NHEJ factors and through histone acetylation, which promotes nucleosome eviction and stimulates resection and subsequent HR events (Figure 6).

In mammalian cells, TIP60 depletion causes defects in DSB repair [152,153,154]. In addition, detailed studies on the effect of individual lysine acetylation are available (recently reviewed in [21]). TIP60-dependent H2A K15 acetylation regulates DSB repair pathway choice by inhibiting H2A K15 ubiquitylation and the binding of 53BP1, thus promoting HR [101]. Acetylation of H4 K5 and H4 K8 by TIP60 facilitates the recruitment of MDC1, BRCA1, 53BP1 and RAD51 [91,155], while acetylation of H4 K16 facilitates both NHEJ and HR [142,154]. Finally, H4 K12 acetylation by the HAT P300/CBP improves the recruitment of SWI/SNF and KU70/80 complexes and promotes the repair by NHEJ [21].

### 7.2. H3 K56

In *S. cerevisiae,* defects in the acetylation of histones that are part of nucleosomes reconstituted during replication are associated with sensitivity to genotoxic agents [156,157]. Cells impaired in H3 K56 acetylation (carrying H3 K56R mutation, or deletions of the nucleosome assembly factor Asf1 or of the H3 K56-specific HAT Rtt109) are extremely sensitive to genotoxic agents, but the origin of this phenotype is still not completely clarified [158,159,160,161,162,163,164]. H3 K56 acetylation facilitates replication-coupled chromatin assembly by increasing the association of new histone molecules with histone chaperones CAF-1 and Rtt106 [157]. Thus, it has been proposed that the sensitivity to genotoxic agents of cells lacking H3 K56 acetylation might result from their defect in chromatin assembly after DSB repair [157,164,165]. This cannot be the only explanation, since strains defective in H3 K56 acetylation are clearly more sensitive to genotoxic agents than chromatin assembly mutants [157]. Moreover, exposure to genotoxic agents triggers Mec1-dependent proteolysis of Hst3 HDAC, which leads to the persistence of H3 K56 acetylation in chromatin after DNA replication [159,166]. Since overproduction of Hst3 causes genotoxic-agents sensitivity [167], its DNA damage-induced degradation and the consequent retention of H3 K56 acetylation is crucial to increase DNA repair processes. Indeed, H3 K56 acetylation reduces nucleosome stability and enhances the rate of DNA end dissociation from nucleosomes, which may facilitate the action of repair activities at DSBs [168,169].

H3 K56 acetylation promotes cell survival after transient exposure to genotoxic agents that cause DNA damage during replication [170]. When H3 K56 acetylation is impaired, exposure to genotoxic agents markedly delays the completion of DNA replication and leads to persistent lesions, bound by the Rad52 HR protein [170]. These data suggest that H3 K56 acetylation in nascent chromatin is important for cells to complete the repair of DNA lesions occurring during DNA replication.

In the mammalian system, both reduction and increase of H3 K56 acetylation have been observed after DNA damage. Deacetylation by Sirt6 and Sirt3 HDACs promotes NHEJ by recruiting the chromatin remodeling enzyme SNF2H and 53BP1 to the DSB sites [91,171].

## 8. Conclusions and Perspectives

Chromatin around a DSB undergoes extensive changes that are regulated at multiple levels to allow DSB repair. Histone PTMs play a pivotal role in the modification of chromatin structure and in the subsequent DSB repair, as highlighted by the hypersensitivity to DSB-inducing agents of cells lacking histone modifier enzymes. In mammalian cells, the regulatory network of histone PTMs involved in DSB response is very complex due to redundancy of the modification machinery and to the variability of the modifications in different cell types. From this point of view, the yeast system has the advantage of a simplified histone PTM machinery. Many components and effectors of this machinery, as well as the biological processes controlled by histone PTMs, are conserved between yeast and vertebrates. Therefore, deciphering the chromatin landscape around a DSB and the effects of histone PTMs on DSB repair pathway choice in yeast should help unveil how these systems operate in mammals.

Although inadequate DSB repair is one of the main causes of genomic instability and tumorigenesis, it also offers therapeutic opportunities to selectively kill cancer cells. In this context, targeting histone modifications required for DSB repair is expected to increase the hypersensitivity of cancer cells with defects in other key factors involved in the DSB response. A comprehensive view of histone PTMs in the surroundings of DSBs and their effects on DSB repair pathway choice is needed both to elucidate the contribution of histone modifications in controlling DNA metabolism and to predict the possible effects of targeting histone modifiers in therapy.

## Figures and Tables

**Figure 1 ijms-24-03248-f001:**
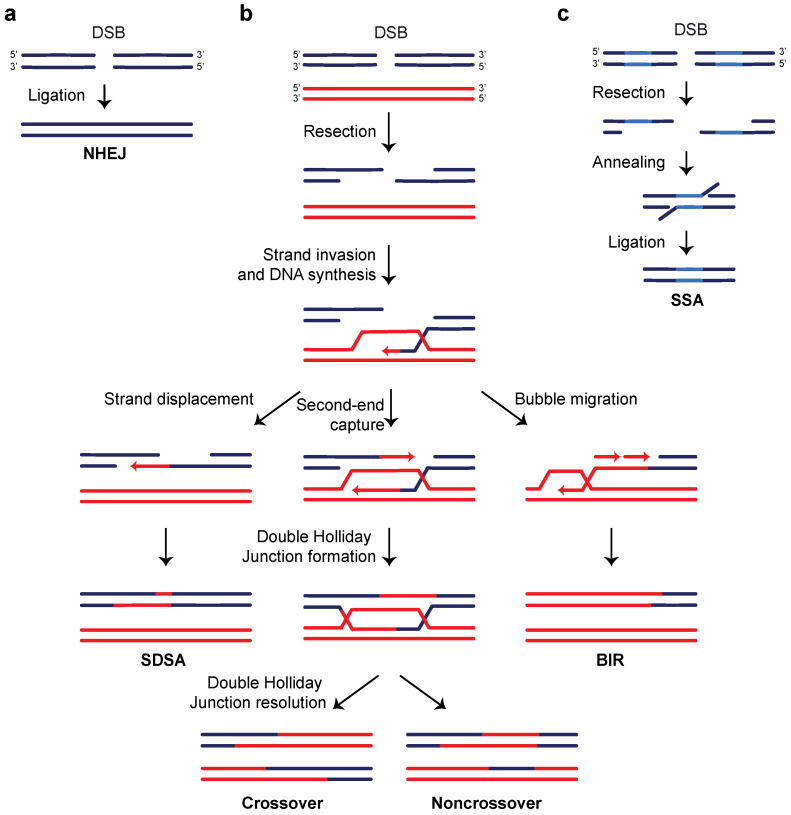
Mechanisms of DSB repair. (**a**) Non-homologous end joining (NHEJ) directly rejoins the two broken ends together. (**b**) Resection generates a 3′-ended ssDNA tail (in dark blue) that invades a homologous duplex (in red) and stimulates DNA synthesis. If the elongating strand re-anneals with the broken template, DSB is repaired by synthesis-dependent strand annealing (SDSA), thus generating only non-crossover products (**left**). If the intermediate is stabilized by a second-end capture, a double Holliday junction is formed. The resolution of this intermediate generates both crossover and non-crossover products (**center**). A failure to engage the second DSB end leads to break-induced replication (BIR). The 3′ overhang that has invaded the homologous template triggers bubble migration and extensive DNA synthesis that can proceed until the end of the chromosome (**right**). (**c**) A DSB between two homologous sequences (in light blue) can be repaired by single-strand annealing (SSA). When resection generates ssDNA at the homologous sequences, they can anneal to each other. Subsequent ligation causes the deletion of one of the homologous sequences and of the intervening sequence.

**Figure 2 ijms-24-03248-f002:**
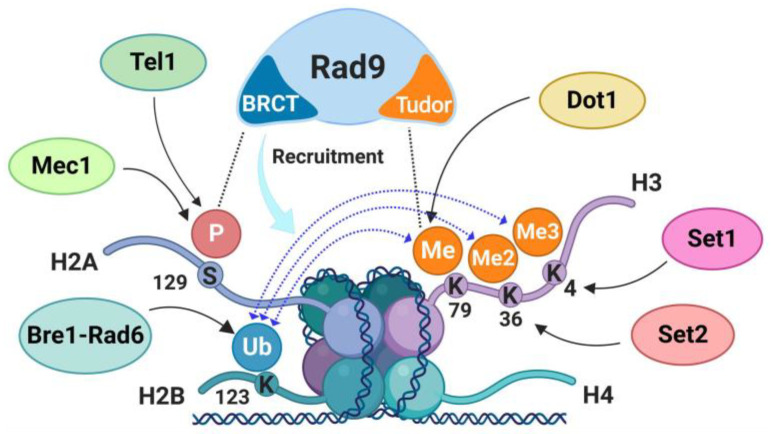
Histone modifications and Rad9 recruitment to DSBs in *S. cerevisiae*. Rad9 is a master regulator of DSB repair pathway choice and its binding to chromatin is regulated by specific histone modifications, comprising H2A phosphorylation by Tel1 and Mec1, H2B ubiquitylation by Bre1-Rad6 complex and H3 methylation by Set1, Set2, and Dot1. Histone PTMs can generate a docking site for Rad9, which binds to phosphorylated H2A S129 residue (γ-H2A) through its BRCT motif and to methylated H3 K79 residue through its Tudor domain. Di- and tri-methylation of H3 K4 and K36 residues likely stabilize the interaction between Rad9 and the chromatin by crosstalk with H2B K123 ubiquitylation. Dotted lines represent molecular interactions with histone marks, dotted lines with arrowheads correspond to epigenetic crosstalk. Created with BioRender.com.

**Figure 3 ijms-24-03248-f003:**
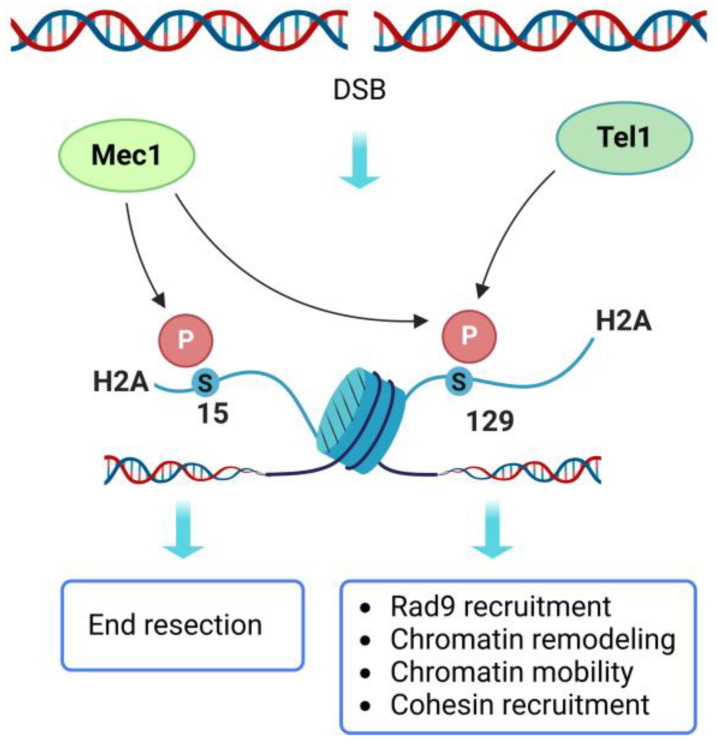
Phosphorylation of histone H2A promotes HR in *S. cerevisiae*. Tel1- and Mec1-dependent phosphorylation of H2A S129 (γ-H2A) regulates both negatively and positively DNA end resection by recruiting Rad9 and chromatin remodeling enzymes, respectively. γ-H2A also increases chromosome mobility, thus favoring homology search, and promotes cohesin recruitment to the DSB ends. Mec1-dependent phosphorylation of H2A S15 stimulates DNA end resection and promotes HR repair. Created with BioRender.com.

**Figure 4 ijms-24-03248-f004:**
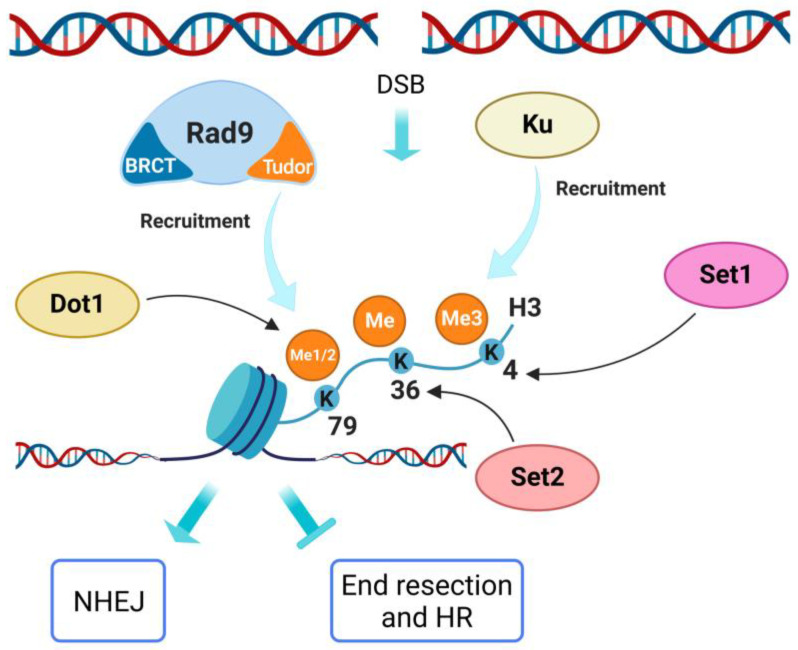
Methylation of histone H3 promotes NHEJ in *S. cerevisiae*. Set1-dependent tri-methylation of H3 K4 residue stimulates an efficient recruitment of Ku to the DSB ends. Set2-dependent methylation of H3 K36 likely limits DNA end resection and chromatin accessibility. Dot1-dependent methylation of H3 K79 is crucial for Rad9 recruitment, which inhibits DSB end resection. Created with BioRender.com.

**Figure 5 ijms-24-03248-f005:**
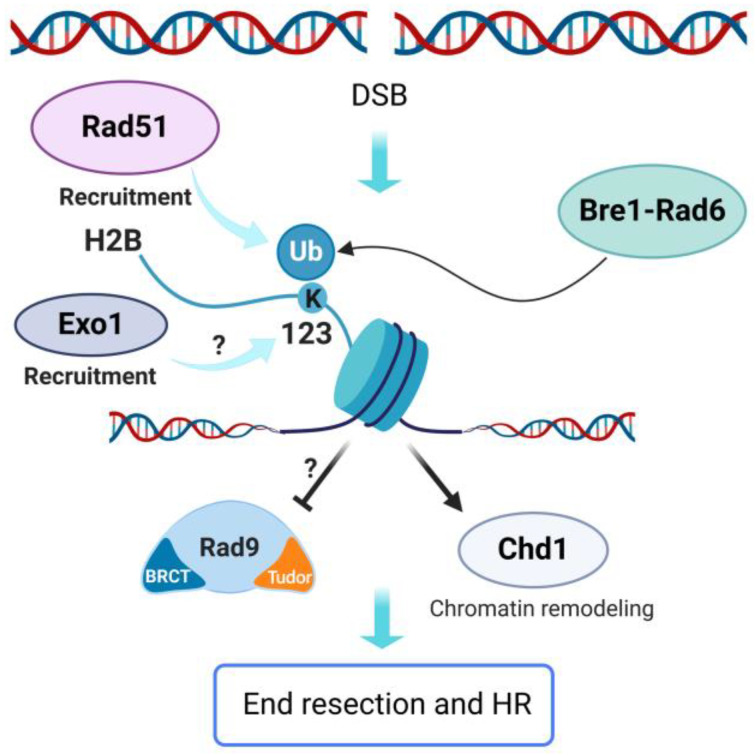
Ubiquitylation of histone H2B promotes HR in *S. cerevisiae*. Bre1-Rad6-dependent ubiquitylation of H2B K123 supports Rad51 loading on resected ends. It also regulates chromatin remodelers like Chd1, thus promoting chromatin accessibility and resection. Finally, H2B K123 ubiquitylation could promote resection by inhibiting Rad9 recruitment and increasing Exo1 association to the DSB ends. Created with BioRender.com.

**Figure 6 ijms-24-03248-f006:**
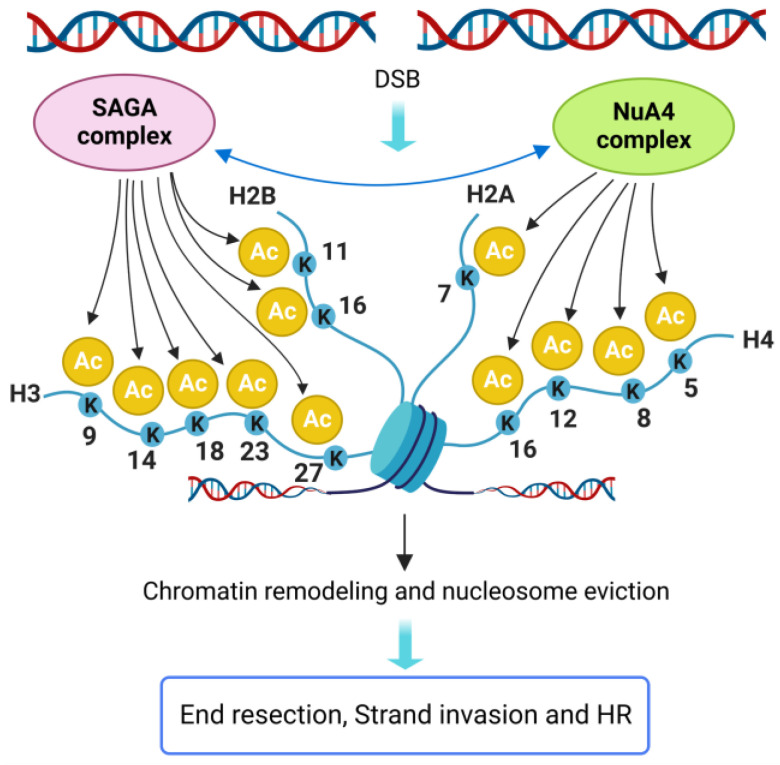
Histone acetylation promotes HR in *S. cerevisiae*. Multiple lysine residues of H2A and H4 histones are acetylated by the NuA4 complex, whose recruitment to the DSB ends is supported by the SAGA complex. SAGA, in turn, acetylates multiple H2B and H3 lysine residues. Histone acetylation causes extensive chromatin remodeling and nucleosome eviction, which promotes both DNA end resection and strand invasion. Double arrowheads line represents protein molecular interaction. Created with BioRender.com.

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
