# Peer review of "The Chromatin Landscape around DNA Double-Strand Breaks in Yeast and Its Influence on DNA Repair Pathway Choice"

_ijms, 2023, doi:10.3390/ijms24043248_

Round 1

Reviewer 1 Report

Frigerio et al reviewed the DSB repair pathway in yeast with emphasis on histone modifications.  The authors describe the DSB repair pathways and the choice for different pathways as well as the chromatin organizations that affect this choice at length.  Despite the lengthy description of histone modifications, there are little to no diagrams to help readers understand the importance of this.  The authors should add more succinct illustrations to help readers comprehend this field.  There are a number of instances where the grammar need to be double-checked including a broken sentence (e.g., 'Therefore, ... in yeast should ...' in conclusion section.  The authors need to go through the manuscript and address these shortcomings. 

Author Response

We thank this referee for its helpful suggestions, and we apologize for the grammar mistakes and the truncated sentence (likely a typo from the last round of revision). We have modified the manuscript according to the referees’ comments. In the revised version of the manuscript we add figures (Figures 3-6) depicting the different classes of post-translational modifications on specific histone residues and accompanying the paragraphs where the corresponding modifications are described. We correct several sentences and double-checked the grammar.

Reviewer 2 Report

Review

ijms-2150980

“The chromatin landscape around DNA double-strand breaks in yeast and its influence on DNA repair pathways choice”

Frigerio et al.

 Dear Authors,

In the following review, the authors have summarized the information about the control of the repair pathways selection via modifications in the chromatin landscape around DSBs. One small drawback is that the review evaluates the influence of chromatin modifications in DSB repair in yeast. However, in many paragraphs, information about the parallel mechanisms in higher eukaryotes is also provided. I would like to suggest, though, increasing these paragraphs as much as possible.

The review article submitted for publication in the journal is very well written, and it will be of significant interest in the field. Therefore, I am experiencing no doubts to suggest the following article for publication in the Journal of Molecular Sciences.

Author Response

We thank this referee for its helpful evaluation and suggestions. When we planned this review, we decided to focus on the chromatin landscape at DSB in yeast because we found that a comprehensive description of these PTMs and their functions in DSB repair was lacking in the current literature, while several recent reviews exist that summarize the mechanisms occurring in mammals. According to the referees’ comments, in the revised version of the manuscript we increase the description of histone PTMs and their effects on DSB pathway choice in mammals. However, the description of pathways operating in mammals is still limited to the space limitation. Furthermore, in the Introduction we refer readers to several recent reviews describing the role of histone PTMs in DNA response in mammals.

Reviewer 3 Report

Clerici et al. discuss the role of the chromatin landscape in the vicinity of double strand breaks (DSBs), especially in terms of post-translational modifications (PTMs) of histones, in the regulation of their repair. In particular, a variety of histone PTMs and protein factors influence the choice between the two main pathways for repair of DSBs: non-homologous end-joining (NHEJ, which can be error-prone) and homologous recombination (HR).

The main objections I find is that the figures do not properly serve to clarify the text, and that the differences between the yeast system, where they focus, and the mammalian machinery, critically concerning PTMs, regulatory proteins, etc, are not clearly enough presented, resulting in somewhat confusing aspects of the manuscript.

Main issues:

1. CORRELATION BETWEEN CELL CYCLE AND CHOICE OF REPAIR PATHWAY. I think that the choice between the two DSBs repair pathways also depends critically upon the cell cycle phase, e.g., HR is only possible during certain phases, and this should be explained somewhere in the Introduction or even the Abstract. By the way, the DSBs repair in the context of stalled replication forks should be mentioned.

2. LACK OF PARALLELISM BETWEEN FIGURES AND TEXT. Figure 1: The correlation between this figure and the text is somewhat confusing: it should be clarified whether both B and C sections correspond to HR. Or could B illustrate also alt-NHEJ? The BIR mechanism is a special case of either Holiday Junction mediated HR or also SDSA? Figures 2 and 3. A similar problem occurs in these figures: Some of the proteins represented in the figure (like Rph1, Gls1) are not mentioned in the text. Neither is mentioned the Tudor domain as a region of Rad9. The contact between Tudor domain and H4K16 mentioned in the legend does not correspond to the line shown in Fig. 2A. The position of residues 4, 36 and 79 on H3 tail makes no logic (with the most terminal depicted more central to the protein). On the other hand, only ubiquitylation of certain residues discussed in the text is shown in the figure, but not others. Finally, the size of text and details in these figures is too small. In summary, the images should be improved to better illustrate / support what is explained in the text.

3. PTMs IN YEAST vs. MAMMALS. Lane 267: “Phospahte groups are added ... on threonine, tyrosine and histidine”. I guess this is an error and you mean serine instead of histidine. If you mean histidine, whose phosphorylation has been rarely reported, at least in eukaryotes, please justify it with bibliographic references. Importantly, protein PTMs might be different in yeast and mammals, the two systems you discuss, and this is crucial for the regulation of repair you are describing. The comparison, either equivalence or differences, between yeast and mammal proteins, (PTMs, target residues, etc) should be more clearly stated in Figures 2 and 3 and through the text. E.g. the PTMs positions you indicate in the figures correspond to yeast or human sequences? In fact, a further figure with a scheme of all the PTMs discussed on every protein sequence (indicating clearly whether yeast or human, or both) should be advisable, since the information is so cumbersome.

4. Recent contributions on chromatin accessibility for repair. I find a recent article by B. Freudenthal, describing the structure of APE1 repair protein sitting on a nucleosome with a lesion-containing DNA as a milestone in these studies: Weaver et al. Nat Commun. 2022 Sep 14;13(1):5390. doi: 10.1038/s41467-022-33057-7. I think this article should be cited. The nice review Scully R et al. Nat Rev Mol Cell Biol. 2019 Nov;20(11):698-714. doi: 10.1038/s41580-019-0152-0, on the choice between the two DSBs repair pathways should be also cited.

Minor issues:

5. Abstract. Lane 14: “... requires several...” (remove “the”)

6. Lane 37. “HR is generally error-free”: right, but it may have also drawbacks, leading to gene deletion of amplification. This might be mentioned.

7. Lane 44 “The DSB repair pathway decision is primarily made at the step of DNA end resection...” I find this sentence confusing, since I guess when resection takes place, the decision is already made...? It might be rewritten.

8. Introduction: Is there any quantitative estimation of the contribution of each route in a given cell? It would be interesting.

9. Lane 110: Please mention what “c-“ states for: “classical”? Lane 112: I suggest giving also the other name: “Microhomology mediated...” (MMEJ)

10. The sentence of lanes 210-211 is repetitive with that of lanes 197-198.

11. Lane 236: Instead of “among which” “including”...? Instead of “that produce” “responsible for high-order...”? Lane 242: instead of “relies on” “is affected by”?

12. Lane 267 “histidine”? See point 3. Lane 274: Instead of “it supports” “parallels”? Last sentence of this paragraph: Better: “This phosphorylation, along with that of S129...”

13. Section 4.1 Better use a homogeneous nomenclature for histone variants: “H2A.X”, etc...?

14. Lane 336: I find this sentence somewhat surprising, since according to your reasoning one would expect that this mutant would improve HR...? Please justify or rewrite.

15. Lane 504 “The HULC...” This sentence is confusing or wrongly built. Please rewrite.

16. Lane 626: “The regulatory network of histone PTMs ... is extremely variable in different cell types.” This is important and casts doubts on the relevance of the PTMs you discuss. One would expect that it varies further between yeast and mammals... This should be warned beforehand and rewritten with a more positive focus...

17. The last sentence of this paragraph is incomplete.

Author Response

We thank this referee for its helpful suggestions, and we apologize for the grammar mistakes and the truncated sentence (likely a typo from the last round of revision). We correct several sentences and double-checked the grammar.

We have modified the manuscript according to the referees’ comments as follows:

Main issues:

Q1. CORRELATION BETWEEN CELL CYCLE AND CHOICE OF REPAIR PATHWAY. I think that the choice between the two DSBs repair pathways also depends critically upon the cell cycle phase, e.g., HR is only possible during certain phases, and this should be explained somewhere in the Introduction or even the Abstract. By the way, the DSBs repair in the context of stalled replication forks should be mentioned.

A1. We explain in the introduction that the choice between HR and NHEJ is a function of the cell cycle phase. We also briefly describe in the Introduction the activation of DSB repair mechanisms at stalled or broken replication forks.

Q2. LACK OF PARALLELISM BETWEEN FIGURES AND TEXT. Figure 1: The correlation between this figure and the text is somewhat confusing: it should be clarified whether both B and C sections correspond to HR. Or could B illustrate also alt-NHEJ? The BIR mechanism is a special case of either Holiday Junction mediated HR or also SDSA? Figures 2 and 3. A similar problem occurs in these figures: Some of the proteins represented in the figure (like Rph1, Gls1) are not mentioned in the text. Neither is mentioned the Tudor domain as a region of Rad9. The contact between Tudor domain and H4K16 mentioned in the legend does not correspond to the line shown in Fig. 2A. The position of residues 4, 36 and 79 on H3 tail makes no logic (with the most terminal depicted more central to the protein). On the other hand, only ubiquitylation of certain residues discussed in the text is shown in the figure, but not others. Finally, the size of text and details in these figures is too small. In summary, the images should be improved to better illustrate / support what is explained in the text.

A2. We modify all the figures according with the suggestions of this reviewer and reviewer #1.

In the new figure 1 we invert panel B and C and rewrite the text accordingly. We add BIR mechanism in the figure

New figure 2. We modify the figure according with the referee suggestions.

New figures 3-6: we add figures depicting the different hisone PTMs occurring in nucleosomal histones in yeast.

In all the figures we increased the text size

Q3. PTMs IN YEAST vs. MAMMALS. Lane 267: “Phospahte groups are added ... on threonine, tyrosine and histidine”. I guess this is an error and you mean serine instead of histidine. If you mean histidine, whose phosphorylation has been rarely reported, at least in eukaryotes, please justify it with bibliographic references. Importantly, protein PTMs might be different in yeast and mammals, the two systems you discuss, and this is crucial for the regulation of repair you are describing. The comparison, either equivalence or differences, between yeast and mammal proteins, (PTMs, target residues, etc) should be more clearly stated in Figures 2 and 3 and through the text. E.g. the PTMs positions you indicate in the figures correspond to yeast or human sequences? In fact, a further figure with a scheme of all the PTMs discussed on every protein sequence (indicating clearly whether yeast or human, or both) should be advisable, since the information is so cumbersome.

A3. Lane 267. We correct the mistake. Threonine, tyrosine and serine residues are phosphorylated.

In all figures we refer to yeast proteins. We state in the figure legends that the residues modified by PTMs depicted correspond to yeast proteins and removed from figures mammalian proteins/histone PTMs.

We add new figures (figures 3-6) depicting the different types of histone PTMs in yeast.

Q4. Recent contributions on chromatin accessibility for repair. I find a recent article by B. Freudenthal, describing the structure of APE1 repair protein sitting on a nucleosome with a lesion-containing DNA as a milestone in these studies: Weaver et al. Nat Commun. 2022 Sep 14;13(1):5390. doi: 10.1038/s41467-022-33057-7. I think this article should be cited. The nice review Scully R et al. Nat Rev Mol Cell Biol. 2019 Nov;20(11):698-714. doi: 10.1038/s41580-019-0152-0, on the choice between the two DSBs repair pathways should be also cited.

A4. We agree with the referee and we cite the indicated papers.

Minor issues:

Q5. Abstract. Lane 14: “... requires several...” (remove “the”)

A5. We correct the sentence

Q6. Lane 37. “HR is generally error-free”: right, but it may have also drawbacks, leading to gene deletion of amplification. This might be mentioned.

A6. We modify the sentence according to the suggestion.

Q7. Lane 44 “The DSB repair pathway decision is primarily made at the step of DNA end resection...” I find this sentence confusing, since I guess when resection takes place, the decision is already made...? It might be rewritten.

A7. We modify the text according to this suggestion. We specify that resection commits repair by HR and that DSB repair pathway decision depends on the structural features of the DSBs and on the timing of the cell cycle.

Q8. Introduction: Is there any quantitative estimation of the contribution of each route in a given cell? It would be interesting.

A8. We think that a quantitative estimation of the contribution of each DSB repair pathways is tricky because the activation of the different pathways depends on the cell cycle phase and on the cell type in mammals. However, we found some quantitative estimation of the relative contribution of NHEJ and HR on average in mammalian cells and reported this information in the Introduction.

Q9. Lane 110: Please mention what “c-“ states for: “classical”? Lane 112: I suggest giving also the other name: “Microhomology mediated...” (MMEJ)

A9. We modify the text according to this suggestion

Q10. The sentence of lanes 210-211 is repetitive with that of lanes 197-198.

A10. We delete the sentence of lanes 210-211.

Q11. Lane 236: Instead of “among which” “including”...? Instead of “that produce” “responsible for high-order...”? Lane 242: instead of “relies on” “is affected by”?

A11. We modify the sentences according to the suggestion.

Q12. Lane 267 “histidine”? See point 3. Lane 274: Instead of “it supports” “parallels”? Last sentence of this paragraph: Better: “This phosphorylation, along with that of S129...”

A12. We correct the mistake in lane 267 and modify the sentences according to the suggestion.

Q13. Section 4.1 Better use a homogeneous nomenclature for histone variants: “H2A.X”, etc...?

A13. We modify the text according with this suggestion.

Q14. Lane 336: I find this sentence somewhat surprising, since according to your reasoning one would expect that this mutant would improve HR...? Please justify or rewrite.

A14. We agree with the reviewer. In the original paper, the authors focused on NHEJ without accessing HR efficiency. They observe a slight increase in NHEJ, but the cause of this increase was not further investigated. We rewrite the sentence, specifying that this result is indeed surprising, and suggest that one possible explanation could be that the H2A-S129E mutation causes a resection delay, contrary to what was observed in the presence of the H2A-S129A mutation, that accelerates resection kinetics. The delayed resection in H2A-S129E cells could give sufficient time to complete NHEJ events.

Q15. Lane 504 “The HULC...” This sentence is confusing or wrongly built. Please rewrite.

A15. We modify the sentence.

Q16. Lane 626: “The regulatory network of histone PTMs ... is extremely variable in different cell types.” This is important and casts doubts on the relevance of the PTMs you discuss. One would expect that it varies further between yeast and mammals... This should be warned beforehand and rewritten with a more positive focus...

A16. We now discuss the difference between the regulatory network of histone PTMs in yeast and mammals in the Introduction, highlighting that, despite the increased complexity of the mammalian system, many aspects of this regulation are conserved between the two organisms and studies in yeast help to define the network regulating histone PTMs and their effects also in mammals.

Q17. The last sentence of this paragraph is incomplete.

A17. We apologize for this mistake. We complete the sentence.

Round 2

Reviewer 3 Report

Authors have made extensive reediting improving the clarity of the text in my opinion.